# Quantifying the dependence of drop spectrum width on cloud drop number concentration for cloud remote sensing

Matthew D. Lebsock[1], Mikael Witte[1,2]

[1]Jet Propulsion Laboratory, California Institute of Technology, Pasadena, CA, USA
[2]Department of Meteorology, Naval Postgraduate School, Monterey, CA, USA

*Correspondence to*: Matthew D. Lebsock (matthew.d.lebsock@jpl.nasa.gov)

**Abstract.** In-situ measurements of liquid cloud and precipitation drop size distributions from aircraft-mounted probes are used to examine the relationship of the width of drop size distributions to cloud drop number. The width of the size distribution is quantified in terms of the parameter $k=(r_v/r_e)^3$, where $r_v$ is the volume mean radius and $r_e$ is the effective radius of the distributions. We find that on small spatial scales (~100 m), $k$ is positively correlated with cloud drop number. This correlation is robust across a variety of campaigns using different probe technology. A new parameterization of $k$ versus cloud drop number is developed. This new parameterization of $k$ is used in an algorithm to derive cloud drop number in liquid phase clouds using satellite measurements of cloud optical depth and effective radius from the MODIS (Moderate Resolution Imaging Spectroradiometer) sensor on Aqua. This algorithm is compared to the standard approach to derive drop number concentration that assumes a fixed value for $k$. The general tendency of the parameterization is to narrow the distribution of derived number concentration. The new parameterization generally increases the derived number concentration over ocean, where $N$ is low, and decreases it over land, where $N$ is high. Regional biases are as large as 20% with the magnitude of the bias closely tracking the regional mean number concentration. Interestingly, biases are smallest in regions of frequent stratocumulus cloud cover, which are a regime of significant interest for study of the aerosol indirect effect on clouds.

## 1 Introduction

Satellite Visible and ShortWave InfraRed VSWIR imagers provide measurements of cloud optical depth ($\tau$) and effective radius ($r_e$) using the bispectral method (Nakajima and King, 1990). There has been an abundance of literature that proposes a translation of these radiative properties to the microphysical properties, liquid water path ($W$) and cloud drop number ($N$), using an adiabatic cloud model (e.g. Boers, 2006; Bennartz 2007; Grosvenor and Wood; 2014; Bennartz and Rausch, 2017). Following the adiabatic model, the number concentration can be expressed as (Grosvenor et al., 2018)

$$N = \frac{1}{2\pi k} \sqrt{\frac{5 f_{ad} c_w \tau}{Q_{ext} \rho_w r_e^5}}, \tag{1}$$

where $f_{ad}$ is the adiabatic fraction (constrained between 0 and 1), $Q_{ext}$ is the extinction efficiency (commonly approximated as 2 in the geometric scattering limit), $c_w$ is the condensation rate, and $\rho_w$ is the density of liquid water.

The parameter $k$ is defined as

$$k = \left(\frac{r_v}{r_e}\right)^3,$$

(2)

where $r_v$ is the volume mean radius and $r_e$ is the effective radius. For a given assumption regarding the shape of the drop size distribution $k$ can be related to various measures of the droplet spectral width. In particular, many studies evaluate the relative dispersion of the DSD defined as the standard deviation normalized by the mean radius, which can be directly related to $k$. For realistic parameterization of the drop size distribution, $k$ is inversely related to the relative dispersion of the DSD.

Grosvenor et al. (2018) have presented a comprehensive uncertainty analysis of $N$ retrievals that use Equation 1. They suggest a pixel-scale uncertainty of 78%, which is dominated by uncertainty in the retrieved $r_e$ owing to the -5/2 power to which it is raised. Note that the parameter with the second largest power in Equation 1 is $k$, which is raised to the -1 power. To our knowledge, all published algorithms that derive $N$ from $\tau$ and $r_e$ have assumed a fixed value of $k$. The review of Grosvenor et al. (2018) propose $k = 0.8 +/- 0.1$ whereas Bennartz and Rausch (2017) suggest a 20% uncertainty in $k$. This intent of this paper is to revisit the uncertainty in remotely sensed $N$ resulting from the assumption of a constant $k$.

The sign of the corelation between $k$ and $N$ is not clear in the published literature. Some analyses of probe data suggest a negative correlation between $k$ and $N$. For example, Martin et al., (1994) find $k = 0.67$ and $k = 0.88$ for a polluted continental cloud and clean oceanic cloud respectively. Similarly, McFarquhar and Heymsfield (2001) find $k = 0.73$ and $k = 0.83$ for polluted and pristine clouds respectively. Liu and Daum (2002) show data implying a negative $k$-$N$ correlation based on probe data from several field campaigns. Other analysis of probe data show a positive $k$-$N$ correlation. Lu et al. (2007) show a positive $k$-$N$ relationship in DSDs measured during the Marine Stratus/Stratocumulus Experiment. Brenguier et al. (2011) evaluate data from five field experiments, finding no discernible $k$-$N$ correlation while attributing some of the correlations reported by previous studies to instrument measurement and sampling artifacts. In addition to evidence from probe data analyses, retrievals of the droplet spectrum width and number concentration derived from the Research Scanning Polarimeter during the North Atlantic Aerosols and Marine Ecosystems Study show a positive $k$-$N$ correlation (Sinclair et al., 2020).

Some of the disagreement in the reported $k$-$N$ correlation can be explained by the spatio-temporal scale on which it is quantified. For example, Pawlowska et al. (2006) show that the relative dispersion of the DSD tends to decrease with $N$ within a given flight leg (implying a positive $k$-$N$ correlation), however the sign of this relationship reverses when evaluating flight averages. Hu et al. (2021) find no correlation for inter-cloud correlation either within campaigns or across the five campaigns they analysed. However, consistent with Pawloswka et al. (2006), they find when analysing the high-resolution 1 Hz DSDs that intra-cloud relative dispersion decreases with increasing number concentration. Hu et al. (2021) further show that the variance explained in the relative dispersion exceeds 40% up to scales of 10 km and quickly decreases at scales above 30 km.

There is a theoretical basis for the scale dependence of the $k$-$N$ correlation. An analytical model based purely on the condensational growth process (Liu et al., 2006) has demonstrated that for a fixed updraft velocity an increase in CCN concentration leads to an increase in $N$ and an increase in relative dispersion, whereas for a fixed CCN an increase in updraft velocity leads to an increase in $N$ but a decrease in relative dispersion. The former effect is interpreted to be relevant at larger scales (inter-cloud) while the latter effect is interpreted to be relevant at smaller scales (intra-cloud). Data from aircraft penetrations of non-precipitating cumulus have confirmed the positive/negative correlation of updraft velocity with $N/k$ respectively on the intra-cloud scale (Lu et al., 2012). The focus of this paper is on the problem of remote sensing, for which the relevant scale is variable depending on the sensor, but is generally within of the microscales ($< 2$ km). We call attention to the fact that many previous parameterizations of the $k$-$N$ relationship are based on significantly larger spatial and temporal scales, which are likely not relevant to the remote sensing problem.

The above explanations do not consider the role of the collision-coalescence process. Recent observations that incorporate measurement of the radar Doppler skewness suggest that 40-50% of marine boundary layer clouds with liquid water path $< 50$ gm$^{-2}$ contain drizzle drops and that drizzle is ubiquitous for thicker clouds (Zhu et al., 2022), which indicates that the collision-coalescence process cannot be ignored in explanations of drop size distribution broadening. A few studies have examined the effect of precipitation on the estimation of $k$ from probe data. Wood (2000) show a reduction in $k$ when drizzle sized drops are included in the calculation and that the ratio of the drizzle and cloud mode liquid water contents can accurately parameterize this reduction. Ackerman et al. (2000) present data from the Monterey Area Ship Track field project which demonstrate that the correlation between $k$ and $N$ can take either positive or negative sign. They show examples in which the sign of the correlation was negative in a non-precipitating Stratocumulus with relatively large $N$ and positive in precipitating Stratocumulus with relatively low $N$. Large Eddy Simulation (LES) of Stratocumulus coupled with bin microphysics shows that coalescence broadening is a key process in the development of a positive $k$-$N$ correlation and that the $k$-$N$ relationship is highly non-linear with the steepest slope at low values of $N$ in the presence of precipitation (Lu and Seinfeld, 2006).

Two points are evident in the body of literature seeking to parameterize $k$: (1) the spatial scale of the data is a critical determinant of the derived $k$-$N$ relationship, (2) the effects of coalescence broadening have frequently been ignored. This paper specifically addresses these issues in the context of deriving $N$ from passive VSWIR remote sensing data.

## 2 Data and Methods

### 2.1 Probe Data

We evaluate measurements of droplet size distribution from three combinations of cloud microphysical probes spanning five airborne experiments:

1. The Aerosol and Cloud Experiments in the Eastern North Atlantic (ACE-ENA) was a deployment of Atmospheric Radiation Measurement (ARM) Gulfstream-1 (G-1) between June 2017 and February 2018 around the ARM Eastern North Atlantic site (Wang et al., 2022). The G-1 sampled primarily marine stratocumulus within 50 km of the ENA site. The primary sampling strategy was an L-shaped pattern in the horizontal with the vertex at the ARM surface site: one leg parallel to the mean PBL flow and one leg perpendicular. The vertical sampling module consisted of in-cloud level legs at cloud base, mid-cloud and cloud top followed by a sawtooth pattern between cloud top and the inversion layer above. Probe data includes the Fast Cloud Droplet Probe (FCDP) and the 2D-Stereo (2DS) Particle Imaging Probe. The FCDP measures drops of diameter 3-50 µm in bins 1.5-3 µm across (increasing with drop size). The 2DS measures drops $10<d<3000$ µm in $10\,\mu m$ increments. We combine the data from each probe using data from the FCDP (d < 30 µm) and 2DS (d > 30 µm).

2. The Cloud System Evolution over the Trades (CSET; Albrecht et al., 2019) sampled Stratocumulus and Cumulus clouds along Lagrangian trajectories between California and Hawaii with the National Science Foundation Gulfstream-V (G-V). The focus of the campaign was understanding the evolution of cloud, thermodynamic, and aerosol properties along the Stratocumulus to Cumulus transition. Boundary layer sampling modules were composed of a sub-cloud level leg, a cloud base level leg and a sawtooth leg between cloud top and the inversion layer above. The G-V microphysical probes include a Holographic Detector for Clouds (HOLODEC) and a 2D Cloud (2DC) probe. The HOLODEC samples drops $6<d<500$ µm with bins from 4-50 µm width (increasing with drop size). The 2DC samples drops $25<d<1550$ µm in 25 µm increments. We combine the data from each probe using data from the HOLODEC (d < 75 µm) and 2DC (d > 75 µm).

3. The Center for Interdisciplinary Remotely-Piloted Airborne Studies (CIRPAS) Twin Otter is frequently flown in coastal stratocumulus (Sorooshian et al. 2018). Here we examine data from two campaigns flown off the coast of Monterey, CA and one off the coast of Iquique, Chile, respectively: Marine Stratus/Stratocumulus Experiment (MASE; Lu et al. 2007); Physics Of Stratocumulus Top (POST; Witte et al. 2017); and VAMOS Ocean-Cloud-Atmosphere-Lands Study (VOCALS; Zheng et al. 2011). Level legs at cloud base, mid-cloud and cloud top were flown for MASE and VOCALS while POST primarily used sawtooth patterns from 100 m below cloud top to 100 m above. Probe data includes the Phase Doppler Interferometer (PDI) and the Cloud Imaging Probe (CIP). The PDI measures drops $2<d<100$ µm in logarithmically space bins of width $dlog_{10}d=0.0156$. The CIP is essentially identical to the 2DC probe used in CSET and samples drops $25<d<1550$ µm in 25 µm increments. We combine the data from each probe using data from the PDI (d<75 µm) and CIP (d>75 µm).

Figure 1 shows an example of a merged DSD from each of the three probe combinations.

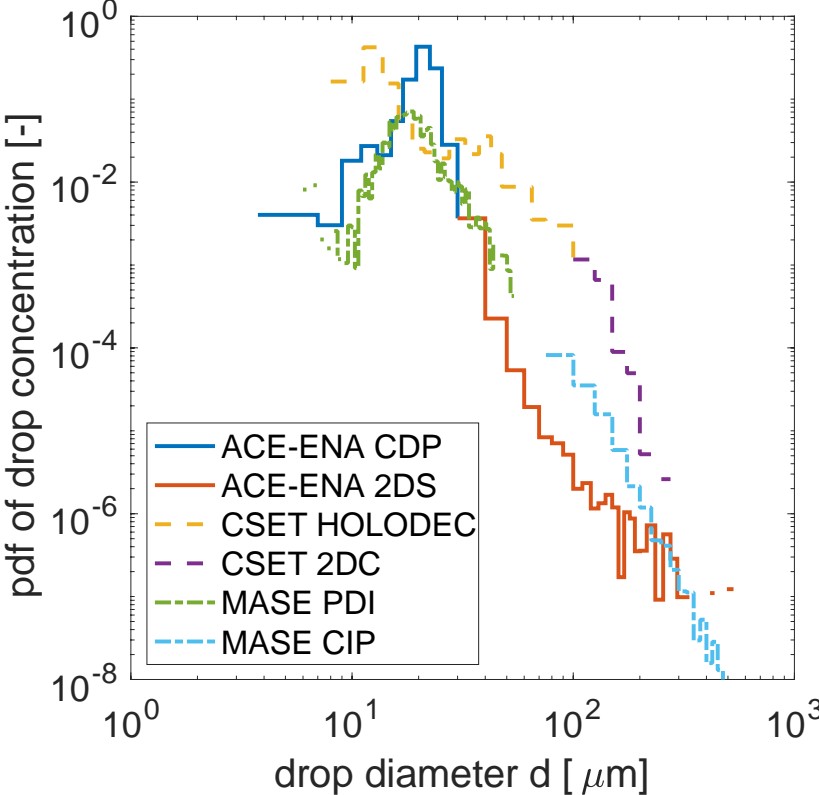

 **Figure 1: An example of merged drop size distributions from each campaign.**

Here we evaluate 1 Hz data, which corresponds to length scale of approximately 100 m. It is acknowledged that the area footprint of a remotely sensed pixel is orders of magnitude larger than the linear sampling provided by the probes. However, it is not possible to reconstruct the two-dimensional area of a typical satellite pixel with probe data. The scale-analysis of Hu et al. (2021) shows that the DSD dispersion relationships at scales less than ~10 km have similar sign, so we choose to evaluate the data at the highest possible resolution and acknowledge that as with other aspects of the remote sensing problem, there is likely to be a pixel heterogeneity bias associated with non-linearities in the *k-N* relationship and the scale of the satellite footprint. For each DSD, several microphysical quantities are calculated. The number concentration is calculated as

$$N = \sum_{r_{min}}^{r_{max}} n(r)\Delta r. \tag{3}$$

Liquid water content is calculated as

$$l = \frac{4}{3}\rho_l \pi \sum_{r_{min}}^{r_{max}} n(r)r^3 \Delta r. \tag{4}$$

The volume mean radius is calculated as

$$r_v = \left(\frac{3l}{4\pi\rho_l N}\right)^{1/3}. \tag{5}$$

The effective radius is defined as

$$140 \quad r_e = \frac{\sum_{r_{min}}^{r_{max}} n(r) r^3 \Delta r}{\sum_{r_{min}}^{r_{max}} n(r) r^2 \Delta r}. \tag{6}$$

Here $n(r)$ is the discretely binned droplet size distribution and $\Delta r$ are the bin variable widths. $r_{min}$ and $r_{max}$ are the bounds of the summation. Each quantity is calculated three times with different min/max values, for the cloud mode, precipitation mode, and total DSD. This requires the definition of an arbitrary threshold radius separating the cloud and precipitation drops, which we choose as 27.5 $\mu$m. The calculation of $k$ is not particularly sensitive to this threshold. For example, Brenguier et al.

(2011) have shown that the calculation of $k$ varies by less than 2% when varying the cloud/precipitation threshold between radii of 25 and 40 $\mu$m, which represents the range that is common to microphysical parameterizations in large eddy models (Geoffroy et al., 2010). The parameter $k$ is further derived from the calculated $r_v$ and $r_e$. In doing so a decision must be made as to whether to include precipitation sized drops in the calculation. In principle, VSWIR imagers see the radiative effects of the entire drop size distribution. However, the SWIR bands from which the measurement of $r_e$ are derived are heavily weighted

towards cloud top (Platnick, 2000). Gravitational settling effectively removes most large drops from the cloud top, therefore here we make the assumption that only the cloud-mode drops ($r < 27.5$ $\mu$m) contribute to the radiatively effective $k$. The assumption that the radiative effects of drizzle mode drops do not contribute to the derived cloud top $r_e$ is consistent with the simulation study of Zhang et al. (2012).

We perform filtering of the data to ensure robust sampling of the cloud mode DSD. We require $l_c > 0.01$ gm$^{-3}$, $N_c > 0.1$ cm$^{-3}$, and at least three of the cloud mode bins have non-zero counts. We further perform additional filtering of the data to ensure that we are sampling cloudy liquid-phase volumes as opposed to sub-cloud drizzle or ice cloud. Specifically, we require relative humidity > 98%, temperature > 273.15 K, and aircraft altitude > 700 m. After filtering there are 157,117 DSDs including 13,205 from CSET, 90,172 DSDs ACE-ENA, and 53,740 from the PDI dataset. Of these 31% have precipitation water content

greater than 0.01 gm$^{-3}$.

## 2.2 Satellite Data

This study uses the collection 6 MODIS Level-2 cloud products from Aqua subset to a 15 pixel (~15 km) swath centered on the CloudSat ground track called MAC06S0 (Savtchenko et al. 2008). These files contain the same data fields as their parent

product, MYD06 (Platnick et al. 2017). Data from 2007-2016 are used. Equation 1 is used to derive the cloud drop number concentration using the effective radius derived from the MODIS band 20, 3.7 $\mu$m channel. Only single layer liquid phase clouds are considered. Cloud phase is determined from the MODIS cloud phase optical properties flag. The parameter $f_{ad}$ is set to 0.66 following Grosvenor et al., 2018 and the condensation rate is expressed as,

$$c_w = \rho_{air} \frac{c_p}{L_v} (\Gamma_d - \Gamma_m) \tag{7}$$

where $\rho_{air}$ is the density of air, $c_p$ is the specific heat of dry air at constant pressure, $L_v$ is the latent heat of vaporization, $\Gamma_d$ is
the dry adiabatic lapse rate, and $\Gamma_m$ is the moist adiabatic lapse rate. Calculation of $c_w$ requires an estimate of cloud base
temperature, which we take as the temperature at the Lifting Condensation Level (LCL) estimated from weather analysis fields
from the ECMWF-aux (Partain and Cronk, 2017) pressure and temperature fields, which is a weather analysis interpolated in
space and time to the CloudSat data. The details of this calculation are provided in Appendix A. Two representations of $k$ are
used; we compare a fixed value of $k = 0.8$ with a parameterization of $k$ based on $N$ derived from probe data in Section 3.1.

## 3 Results

### 3.1 Drop Size Distribution Properties

We begin by showing the distributions of microphysical variables for the three datasets in Figure 2. The range of cloud drop
number spans 0-500 cm$^{-3}$. The PDI dataset is weighted towards higher $N$ while the FCDP and especially the HOLODEC
datasets are weighted towards the lowest $N$. These differences are a consequence of the geographic regimes sampled by the
different campaigns: the CIRPAS Twin Otter is constrained to fly near the coast and the associated natural and anthropogenic
aerosol sources, while much of the boundary layer sampling in CSET took place over extremely clean remote oceanic regions.
The corresponding effective radii distributions range between roughly $5 - 25$ $\mu$m. Cloud liquid water content distributions are
similar across the datasets, with a broad range of values from the artificial cut off at 0.01 gm$^{-3}$ to greater than 1 gm$^{-3}$. The
distributions of $k$ show that the PDI have the largest $k$ followed by the FCDP and then the HOLODEC. The mean value of $k$ is
0.8, which is consistent with the recommended value of Grosvenor et al., 2018. The negatively skewed distribution, with values
below 0.6 not uncommon, is consistent with the multi-angle polarimetric remote sensing derived distributions of $k$ shown in
Grosvenor et al., 2018 (their Figure 12).

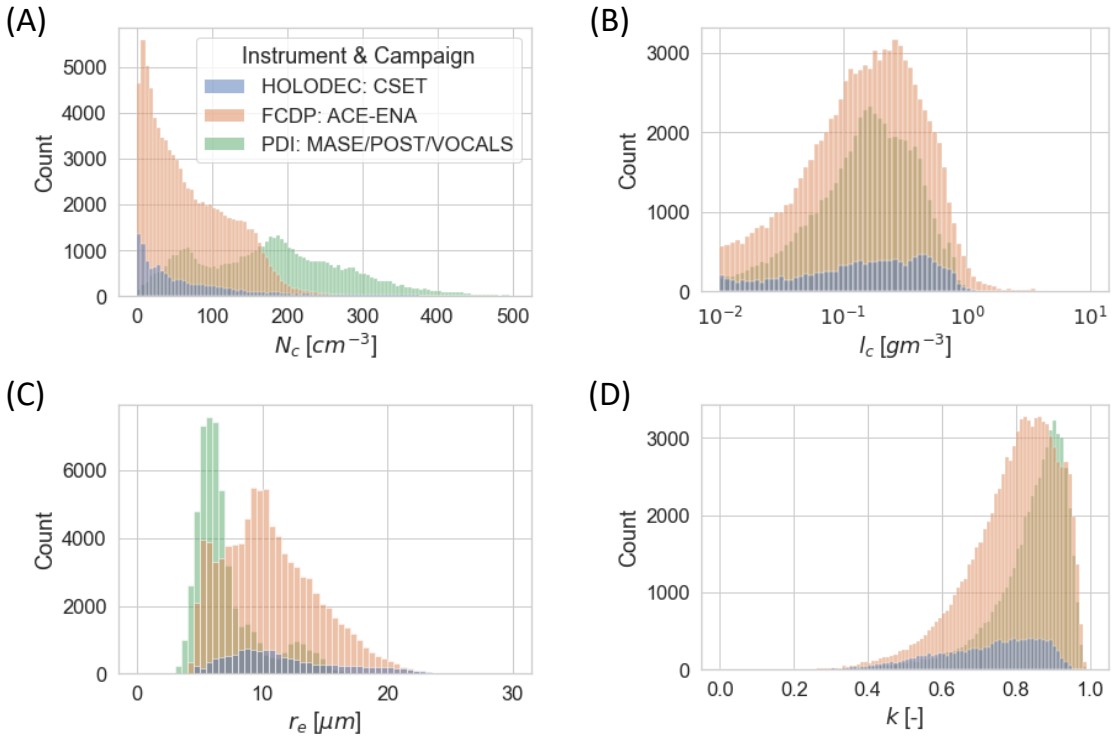

**Figure 2: Distribution of cloud drop number (A), cloud liquid water content (B), effective radius (C), and $k$ (D) for the three datasets.**

We now turn to the relationship between $N$ and $k$. Figure 3 shows the mean DSDs for each dataset sorted by the cloud drop
number concentration. Here we include two filters applied to the data; The first includes all DSDs (Panels A-C), while the
second includes only DSD's that have no water in the precipitation mode (Panels D-F). Note that the calculation of $k$ is only
based on the cloud mode drops ($r < 27.5$ $\mu$m) even for the DSDs that do have precipitation sized drops. Including precipitation
sized drops in the calculation (not shown) results in significantly smaller values of $k$ consistent with Wood (2000), however as
discussed in Section 2.1 we believe that including precipitation sized drops is not appropriate in the context of the remote
sensing application. There are some clear biases in the data. First, the FCDP data from ACE-ENA show consistently low
counts in several bins around 6 μm radius. This minimum is not observed in the DSDs from the other datasets nor is there any
microphysical reason to expect this feature. Second, the CSET DSDs are limited by an inability to sample the smallest drops.
Third, the PDI data shows noisiness in the smallest bins. Despite these artifacts, there is a clear broadening signature as $N$
decreases in all of the DSDs, which use completely different probes and sample different cloud regimes. The broadening is
most obvious in the drizzle sized drops but is also clearly present in the cloud-size drops (Panels A-C). It is also apparent in
the DSDs that have no precipitation water (Panels D-F). Furthermore, these results are robust when controlling for variation
in the cloud liquid water content (Figure S1).

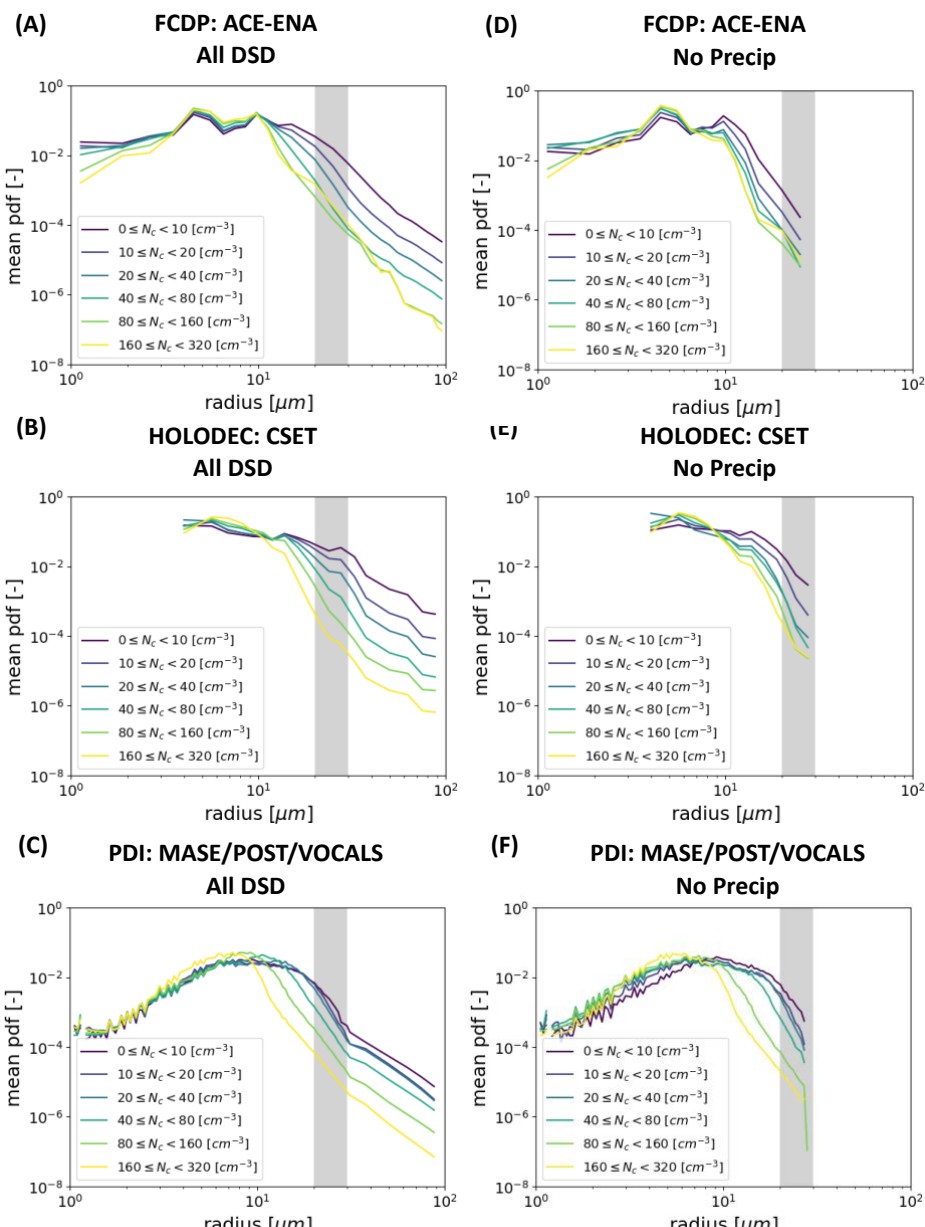

Figure 3: The mean probability density function (pdf) of the drop concentration sorted by the cloud mode drop number concentration for the three datasets. Panels A-C show results for all DSDs and panels D-F shows results filtered for DSDs with no precipitation water. The parameter $k$ is calculated using only the cloud-sized drops even when precipitation is present in the DSD. The grey bar shows a notional range of radius 20-30 $\mu$m separating cloud from precipitation sized drops.

Figure 4 shows the dependence of $k$ on $N$ for each of the three datasets. Panel A shows the results for non-precipitating DSDs, while panel B is filtered for only precipitating DSDs. The increase of $k$ with $N$ is clear across all three datasets. The median values increase from around 0.6 to 0.9 across the range of $N$ from $0 - 500$ cm$^{-3}$. The variability around this mean value is large for small $N$ but decreases consistently as $N$ increases. The general relationship is consistent across all three datasets however

there are some quantitative differences especially for the lowest $N$ where the HOLODEC data have smaller $k$ values than either the FCDP or the PDI data. It is possible that this low bias in HOLODEC results from the inability to sample the smallest drops, however this is not guaranteed. For example, revisiting Figure 3, it is clear that the smallest drops increase the DSD width for the FCDP data but decrease it for the PDI data. More work is warranted to resolve the differences between the probes with regards to the $N$-dependence of the sampling of smallest drops. Returning to Figure 4, the dependence of $k$ on $N$ appears

regardless of the presence of precipitation in the DSDs, however, there is a tendency for $k$ to be larger (narrower DSD) when the precipitating DSDs are removed, which is particularly evident at low values of $N$. This effect is especially evident in the CSET HOLODEC data which happens to be the dataset with the lowest number concentration. This suggests, that although collision-coalescence is not necessary to explain the observed $k$-$N$ correlation it likely plays a role in strengthening that correlation.

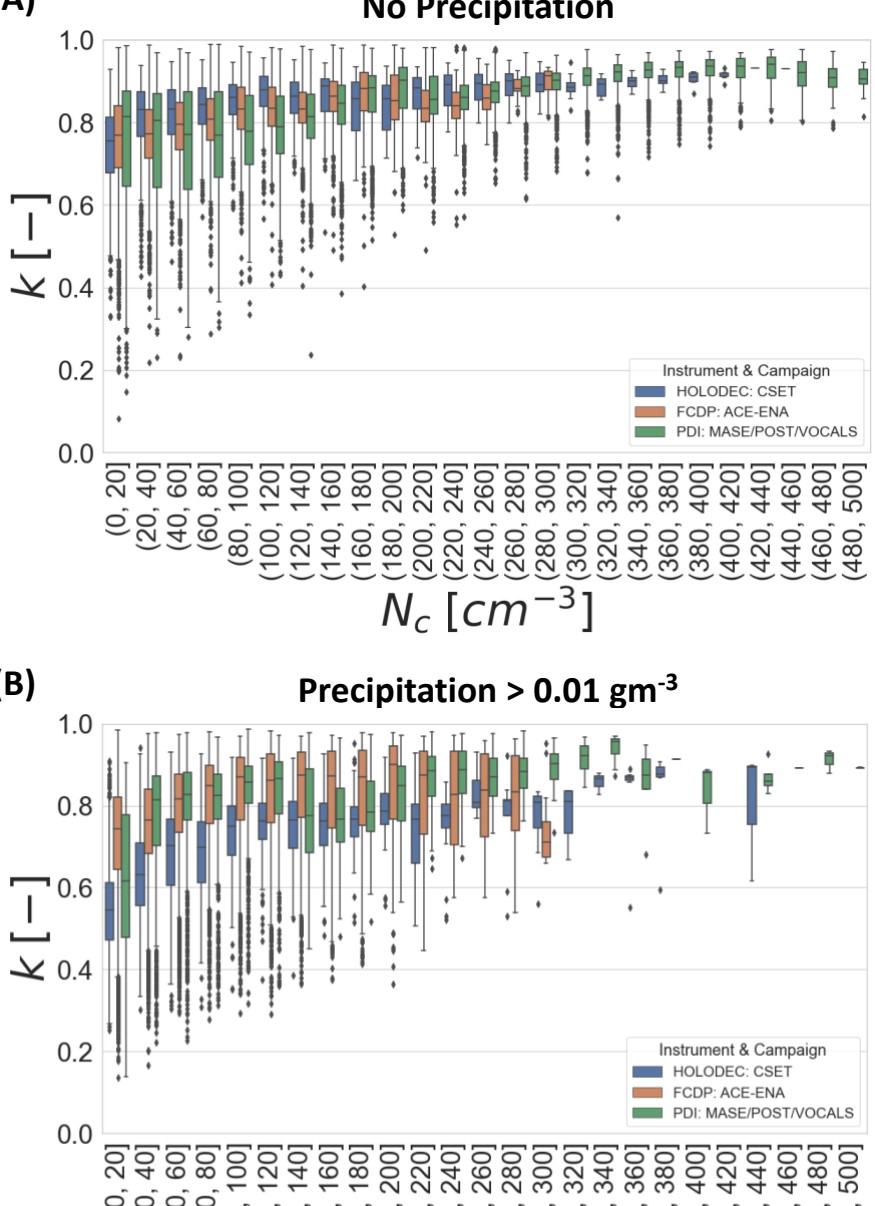

**Figure 4: Box and whisker plot shows the distribution quartiles and outliers of _k_ as a function of _N_ for the three different instruments. Panel A shows results for non-precipitating DSDs and panel B shows results filtered for DSDs with precipitation water. The parameter _k_ is calculated using only the cloud-sized drops even when precipitation is present in the DSD.**

Given these results, we propose the following parameterization for the $N$-dependence of $k$ for clouds

$$k(N) = k_1 + (k_2 - k_1)\left(\frac{N}{N+N^*}\right), \tag{8}$$

with $0 \le k_1 < k_2 \le 1$, $N^* > 0$. This functional form is ad-hoc, however it has two important properties: (1) it is bounded between 0 and 1, which is expected from physically realistic size distributions and (2) it increases monotonically and non-linearly while saturating for larger values of $N$, as we observe in the data. We use the python curvefit package to implement the Levenberg-Marquardt method to perform a non-linear least-squares regression to fit the parameters of Equation 8 for each dataset individually, and for a weighted-combined dataset. Fitting parameters are shown in Table 1. The weighting in the combined data is given by $1/\sqrt{M}$ where $M$ is the sample count for each dataset. In the absence of this weighting the combined fit is nearly identical to the ACE-ENA fit, which has the largest sample size. The raw data and the curve fits are shown in Figure 5. All of the fits show a clear positive correlation between $N$ and $k$, however the PDI and FCDP $k$ values are larger than the CSET values, particularly for the smallest $N$. In contrast the PDI data and the ACE-ENA have larger values of $k$ at large $N$. By design, the combined fit falls in the middle of the distributions. Given the limitations of each of the probe datasets (e.g. Figure 3) and the limited sampling of the datasets used here, we take this combined fit as our best-estimate parameterization of $k(N)$.

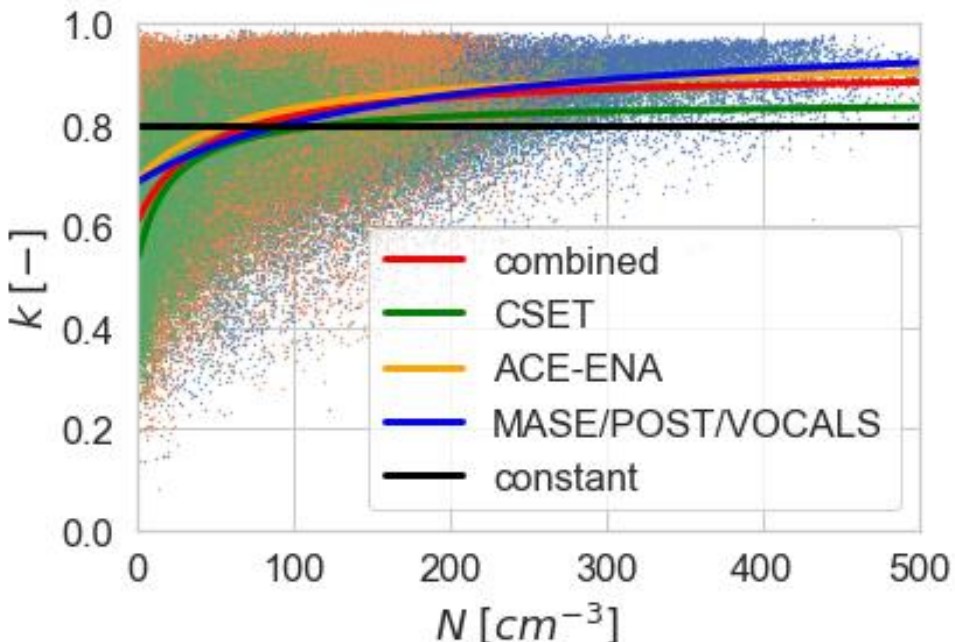

**Figure 5: Curve fits for Equation 7 to the data from each of the three data sets as well as a weighted combined dataset. The weighting normalizes the influence of each of the three datasets due to the dramatically different number of data points in each dataset.**

**Table 1: Fitting parameters for Equation 7.**

|  | $k_1$ | $k_2$ | $N^*$ |
|---|---|---|---|
| Combined | 0.61 | 0.90 | 43 |
| HOLODEC: CSET | 0.53 | 0.85 | 22 |
| FCDP: ACE-ENA | 0.69 | 0.94 | 73 |
| PDI: MASE/POST/VOCALS | 0.68 | 1.0 | 163 |

## 3.2 Satellite Retrievals

Here we incorporate the new parameterization of $k$ into retrievals of $N$. To begin, we write Equation 1 as

$$N = \frac{\phi(\tau, r_e)}{k(N)}. \tag{9}$$

Plugging in Equation 8 for $k(N)$ and rearranging terms results in the quadratic equation

$$k_T N^2 + (k_B N^* - \phi)N - \phi N^* = 0. \tag{10}$$

Employing the quadratic formula and taking the positive root, gives the following solution for $N$,

$$N = \frac{\phi - k_B N^* + \sqrt{(k_B N^* - \phi)^2 + 4k_T \phi N^*}}{2k_T}. \tag{11}$$

Equation 11 is specific to the parametric form of $k(N)$ provided in Equation 8, however we note that if $k$ is an arbitrary monotonic function of $N$, an iterative solution can also be employed to solve for $N$. In the results that follow we use the parameters for the combined fit for $k(N)$ given in Table 1.

Figure 6 shows $N$ as a function of $\tau$ and $r_e$ derived from Equation 11, along with the bias in the derived $N$ caused by ignoring the parameterized $k$-$N$ correlation. The parameterization decreases $N$ for the largest values and increases it for the smallest values. Over the range of $\tau$ and $r_e$ shown the bias varies between roughly -10 % and 30 %. The sign of the bias switches near the value of 81.7 cm$^{-3}$. The co-distributions of the observed MODIS $r_e$ and $\tau$ that are used in this study are also overlaid on these plots. Note in Figure 6B that these observed distributions span the range of positive and negative bias, with a tendency to lie near the zero-bias ridge. A result is that the mean bias in derived $N$ caused by ignoring $k$-$N$ correlation will be small even if it can be significant in the case of the extreme values.

**(A)**                                                  **(B)**

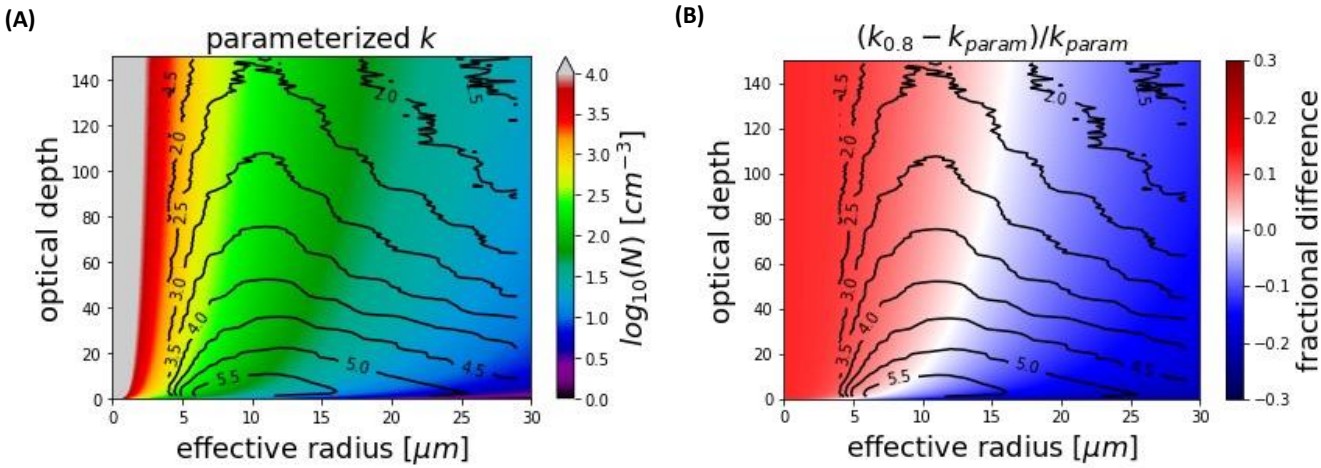

**Figure 6: Derived $N$ using the parameterized $k$ and $c_w = 0.004$ gm$^{-4}$ (A) and bias between the parameterized $k$ and constant $k$ retrievals (B). The black contours show the base 10 logarithm of the counts of the observed MODIS effective radius and optical depth used in the study.**

Figure 7 shows the regional pattern of $N$ derived using the $k$-$N$ parameterization in panel A. Implementing the parameterization makes essentially no difference in the qualitative understanding of the distribution of $N$ globally. The mean $N$ tends to be lowest over remote ocean areas, increasing in regions of low altitude stratocumulus over ocean, and highest over populated regions over land. Nevertheless, there are subtle quantitative differences in the retrieved $N$ when using the parameterized $k$. The Root-Mean-Square-Deviation can be as large as 20% in regions of low mean $N$. Importantly, the RMSD is smallest in the

regions of stratocumulus off the west coasts of the continents and in the mid-latitude storm tracks. This results from the fact that the $k$-$N$ parameterization is near $k = 0.8$ given the mean $N$ in these cloud regimes. This result explains why assuming a constant $k = 0.8$ tends to work reasonably well, since these are the regimes that contribute the bulk of the observed low cloud retrievals by the VSWIR sensors. Finally, the regional distribution of uncertainties are reflected in a bias in the mean which approaches -20% in remote ocean areas and is smaller than about 5% in polluted areas over land.

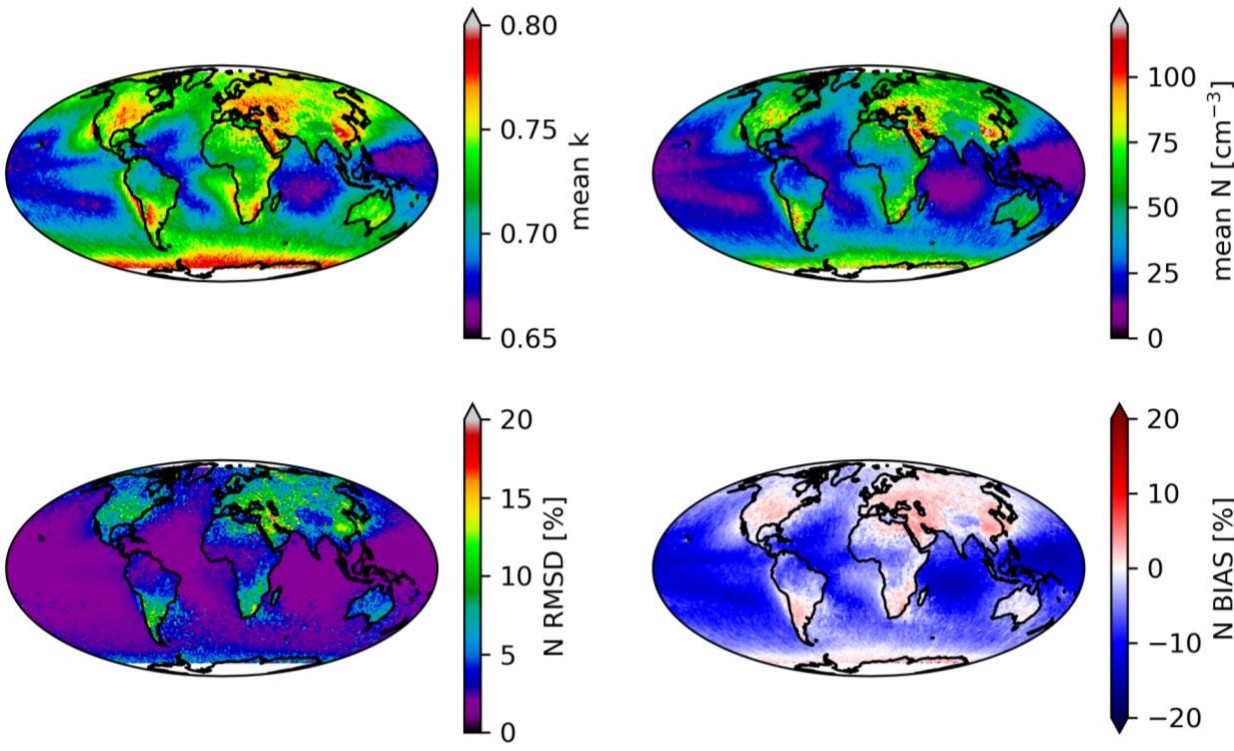

**Figure 7: Mean value of the *k* parameter (top left), Derived *N* using the parameterized *k* (top right), Root-Mean-Square-Deviation (RMSD) between parameterized and constant k retrievals (bottom right), and mean bias between the parameterized *k* and constant *k* retrievals (bottom right). Here Bias is defined as a bias of the constant k retrieval relative to a variable *k* retrieval.**

## 4 Summary and Discussion

We have quantified the dependence of a parameter describing the droplet spectrum width (*k*) on the cloud droplet number concentration (*N*). We specifically address this dependence at the small spatial scales relevant to cloud remote sensing by using 1 Hz data that corresponds to an approximately 100 m spatial scale. We find that, using the high-frequency data, *k* and *N* are positively correlated, which is in agreement with a number of recent studies. This finding is robust across 5 different campaigns employing three different drop measurement technologies. The sign of this correlation is consistent with theoretical

expectations that at the micro scale (<2 km), variation in vertical velocity within a cloud, affect the supersaturation in a manner that induce a positive *k-N* correlation (Liu et al., 2006). We have further shown here that when the drop size distributions are sorted by the presence of precipitation, the distribution of *k* broadens, and the slope of the *k-N* correlation increases. This finding points to the added importance of the collision-coalescence process in droplet spectrum broadening, which is consistent with large eddy simulations coupled to bin microphysics (Lu and Seinfeld, 2006).

Based on these findings, a parameterization of $k$ given $N$ is presented for use in satellite remote sensing of the droplet number concentration. This parameterization can easily be incorporated into the existing techniques to derive $N$ from VSWIR imagery. We show that imposing the positive $k$-$N$ correlation tends to narrow the distribution of retrieved $N$ by increasing the smallest values and decreasing the largest values. The parameterization is applied to retrievals of $N$ using data from Aqua-MODIS. The fractional bias in the remotely sensed number concentration approaches -20% in areas of the remote ocean where number concentrations are low. A smaller positive bias is found over polluted land regions. We do emphasize that the absolute magnitude of the bias in $N$ tends to be relatively small, on the order of 1-2 cm$^{-3}$. Nevertheless, the data are robust in showing that there is a positive $k$-$N$ correlation and employing this dependence in retrievals is relatively straightforward. Therefore, we argue that these findings inform the implementation of future retrievals of drop number concentration from satellite remote sensing. We provide the $k$-$N$ relationship at the highest possible spatial resolution with the understanding that this may introduce bias due to sub-pixel inhomogeneity. Knowledge of the sub-pixel distribution of $N$ could be used to correct any such bias.

While it is straightforward to implement the parameterization presented here for the derivation of $N$ from existing $\tau$ and $r_e$ retrievals, it is important to realize that there is a physical inconsistency between the parameterized $k$ and the $k$ assumed in the original retrieval of $\tau$ and $r_e$. The MODIS retrievals assume a constant value of k = 0.72 (Grosvenor, 2018). It has been shown that bias in the retrieved $r_e$ is inversely related to the bias in $k$ (Zhang, 2013). That is to say that if the assumed $k$ in the $\tau/r_e$ retrieval is too large then the retrieved $r_e$ will be too small. An implication of this work is that the MODIS $r_e$ may contain a bias related to the relationship between $k$ and $N$. Since the average $k$ in the probe data is 0.8, on average this will bias MODIS $r_e$ low and the resultant $N$ high, however the opposite would be true for very low $N$ clouds. One solution to this inconsistency would be to build $\tau/r_e$ look up tables using an adiabatic model with input $N$, cloud thickness, and condensation rates in a manner similar to Schüller et al., (2003), while also including the $k$-$N$ parameterization used in this paper. One could further increase the realism of the look up tables by including a parameterization of the cloud subadiabaticity as a function of cloud thickness as in Schulte et al. (2023). In this manner a complete and physically consistent description of the cloud microphysical profile can be constructed from the initial inversion of the radiances.

The emphasis of this paper is on the $k$-$N$ correlation on the microscale ($\leq$ 2 km), which is relevant to the problem of remote sensing. Many previous studies have focused on the $k$-$N$ correlation at the meso-beta scale (20-200 km) with a science focus on parameterizing the aerosol indirect effect in climate models (e.g. Rostayn and Liu, 2003; Xie et al., 2017). While this study finds a positive $k$-$N$ correlation, most studies exploring the aerosol indirect effect find or assume a negative correlation. Liu et al., 2006 has suggested that the positive microscale correlation is a result of variations in updraft velocity, whereas the negative mesoscale correlation is a result of variation in aerosol number concentration. Therefore, the correlations found here may not

be applicable to evaluation of the aerosol indirect effect in climate simulations. We recommend that more work is necessary to fully resolve the scale-dependence of the *k-N* relationship and its implications for aerosol indirect effects.

## Appendix A

Calculation of the moist adiabatic condensation rate requires an estimate of the difference between the dry and moist adiabatic lapse rates and the density of air. The dry adiabatic lapse rate is,

$$\Gamma_d = \frac{g}{c_p} \tag{A1}$$

where $g$ is the gravitational acceleration and $c_p$ is the specific heat at constant pressure of dry air. The moist adiabatic lapse rate is given by,

$$\Gamma_m = g \frac{1 + \frac{L_v r_v}{R_d T}}{c_p + \frac{L_v^2 r_v \epsilon}{R_d T^2}} \tag{A2}$$

where $L_v$ is the latent heat of vaporization, $r_v$ is the mixing ratio of water vapor, $T$ is the temperature, and $R_d$ is the gas constant for dry air. To include the temperature dependence in gamma we estimate cloud-base temperature as the temperature at the LCL. We calculate the LCL height ($Z_{LCL}$) following Bolton (1980) from the surface relative humidity (*RH*)

$$Z_{LCL} = \frac{1}{\Gamma_d}\left(T - 55 - \left(\frac{1}{T-55} - \frac{\ln(RH)}{2840}\right)^{-1}\right) \tag{A3}$$

where the *RH* varies between 0 and 1 and the *T* is in K. The surface *RH* is calculated from the ECMWF specific humidity ($q$)

$$RH = \frac{q}{q_{sat}} \tag{A4}$$

and the saturation specific humidity ($q_{sat}$) is calculated from the saturation water vapor mixing ratio ($r_{sat}$)

$$q_{sat} = \frac{r_{sat}}{1 + r_{sat}} \tag{A5}$$

which in turn is calculated from the saturation vapor pressure ($e_{sat}$) and the surface Pressure (*P*)

$$r_{sat} = \epsilon \frac{e_{sat}}{P - e_{sat}} \tag{A6}$$

where $\epsilon = 0.622$. The saturation vapor pressure is calculated following the Tetens formula

$$e_{sat} = 6.1078 e^{\frac{17.269388T}{T+237.3}} \tag{A7}$$

Where *T* is degrees Celsius and $e_{sat}$ in mb. The temperature at the LCL is calculated assuming a dry adiabatic lapse rate using the ECMWF 2-meter temperature

$$T_{LCL} = T_{2m} - \Gamma_d Z_{LCL} \tag{A8}$$

The pressure at the LCL is calculated using the surface pressure and a scale height ($Z^* = 8\ km$)

$$P_{LCL} = P_0 e^{-Z_{LCL}/Z^*} \tag{A9}$$

The air density at the LCL is calculated

$$\rho_{air,LCL} = \frac{P_{LCL}}{R_d T_{LCL}} \tag{A10}$$

Finally, the water vapor mixing ratio at the LCL, needed in A2, is assumed to be the saturated mixing ratio and is calculated A6 and A7 using the $T_{LCL}$ and $P_{LCL}$.

**Code/Data Availability**

*CloudSat* and MODIS data used in this study were downloaded from the CloudSat Data Processing Center
(http://www.cloudsat.cira.colostate.edu/). ACE-ENA data is available from https://adc.arm.gov/discovery/#/. CSET data is available from https://data.eol.ucar.edu/master_lists/generated/cset/. MASE data is available from https://zenodo.org/records/1035928. POST data is available from https://data.eol.ucar.edu/dataset/list?project=96&children=project.  VOCALS data is available from https://data.eol.ucar.edu/dataset/list?project=215&children=project.

**Author Contribution**

M.W. compiled the drop size distribution data and merged the cloud and precipitation distributions. M.L. evaluated the drop size distributions, implemented the retrieval, and prepared the manuscript.

**Competing Interests**

The authors have the following competing interests: At least one of the (co-)authors is a member of the editorial board of Atmospheric Chemistry and Physics.

**Acknowledgments**

This work was performed at the Jet Propulsion Laboratory, California Institute of Technology, under a contract with the
390 National Aeronautics and Space Administration and was funded by the *CloudSat* mission.

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
