# Peer review of "Quantifying the dependence of drop spectrum width on cloud drop number concentration for cloud remote sensing"

_EGUsphere, 2023_

## Referee Comment (RC2)

**Review of Lebsock and Witte, 2023.**

This paper examines the dependence of the k-value (k=(rv/re)^3, where rv is volume mean radius and re is the effective radius) of cloud droplet size distributions upon the cloud droplet concentration (Nd) using aircraft data. A parameterization is developed to describe the dependency. This dependency leads to biases in the retrieval of Nd using MODIS retrieved re and cloud optical depth using the adiabatic cloud assumption since a constant k value is generally assumed for such retrievals. The biases due to assuming a constant k value of 0.8 are quantified using MODIS data subset to the CloudSat ground track (MAC06S0 dataset), presumably at ~1km resolution (although this is not clear from the text – see comments below).

The science seems sound and the paper is very well written. The only thing missing is a little more discussion on how this might combine with some of the other potential biases relating to the precipitation, the k value and associated droplet distribution widths, and some discussion on the implications for climate models (which relates to the behaviour at larger spatial scales).

**Major points**

It would be good to discuss the effect of k variability and the presence of precipitation on the MODIS retrievals of re. MODIS assumes a modified gamma distribution for cloud droplets for the radiative models used for its retrievals of re. The variation of k will also affect this and may lead to biases in re that would be likely to have a strong influence on Nd retrievals. The presence of a precipitation mode might also affect the retrieved re and hence Nd retrievals. I don't expect you to investigate the effect of this, but it would be worth mentioning in the discussion perhaps. Correcting for this bias could therefore either enhance or reduce the bias discussed in the paper being reviewed. Section 2.4.4. of Grosvenor et al. (2018) contains some useful discussion on this. From the information given there it seems likely that in precipitating regions neglecting the rain mode is likely to lead to an underestimate of Nd. These are likely to be regions with broader droplet distributions (low k) and the presence of a precipitation mode may therefore enhance the error seen in your paper since you also show an underestimate of Nd in such regions. In regions of high k the assumption by the MODIS retrieval of k=0.72 is likely to lead to an underestimate of Nd due to the DSD width assumption in the re retrieval, whereas your paper shows a positive Nd bias in such regions.

Zhang, Z. B., Ackerman, A. S., Feingold, G., Platnick, S., Pincus, R., & Xue, H. W. (2012). Effects of cloud horizontal inhomogeneity and drizzle on remote sensing of cloud droplet effective radius: Case studies based on large-eddy simulations. Journal of Geophysical Research, 117, D19208. https://doi.org/10.1029/2012JD017655

Zhang, Z. (2013). On the sensitivity of cloud effective radius retrieval based on spectral method to bi-modal droplet size distribution: A semi-analytical model. Journal of Quantitative Spectroscopy & Radiative Transfer, 129, 79–88. https://doi.org/10.1016/j.jqsrt.2013.05.033

These papers might be useful for the discussion of drizzle effects too at line 70.

Some climate models use the parameterization of Liu (Y. Liu, P. H. Daum, H. Guo, and Y. Peng. Dispersion bias, dispersion effect and the aerosol-cloud conundrum. Environ. Res. Lett., 3:045021, 2008.), and other similar papers from those authours. For example the UK Earth System Model (UKESM) as described in Mulcahy (2018;

https://agupubs.onlinelibrary.wiley.com/doi/epdf/10.1029/2018MS001464). It would be good to discuss the fact that this could also be important for climate models in the introduction. It would also be useful to discuss what the implications of the results of your paper are for such climate models – this would probably relate to the issue of the scale dependence. The Liu and Daum parameterization has the opposite dependence of k on N to your parameterization, although the scales of a climate model grid box are much larger (1x1 degree or so, so around 100km). The impact may depend on the details of how SW radiative fluxes are calculated in the climate models (e.g., using grid-box wide values or sub columns, etc.).

**Minor points**

Eqns.3 to 6 – you should probably make it clear that you are summing over the different size bins of the droplet probes with subscript i values for the bin quantities (n(r), r, $\Delta$r) and you should say what n(r), rmin and rmax refer to. Also, Eqn. 6 switches to using a capital N for n(r).

L146 – "15 pixel (± 5 km) swath" – this is a bit confusing since MODIS pixels for re at least are usually 1km across. Should it be 10 pixels, or ± 7.5 km? Otherwise it would be good to explain this in the text. You should also say what the resolution of the MODIS data that you are using is.

L225 – since you reduce the weighting of the campaigns with more sample points in order to get more equal representation between the campaigns, would a straight average of the three curves not achieve something similar?

Fig. 5 – it seems like a 2d histogram would be better than the scatter points (or contours with different colours for each campaign if wanting to separate them). You should also say what the scatter points are in the caption.

**Typos**

L45 – Heymfield -> Heymsfield

L69 - that collision coalescence process -> that the collision coalescence process

l79 - at low values N -> at low values of N

L86 - measurements of drops size distribution -> measurements of droplet size distributions

L268  - difference -> differences

---

## Author Comment (AC1)

Thank you to both reviewers for your time and effort. The reviewer's efforts have definitely improved the manuscript. Below, the reviewer comments are in black text, our response is in blue text, and modified test in the manuscript is in red italics. We believe that our revised manuscript addresses the reviewer's comments. We have accepted most of the reviewer's suggestions except for reviewer #1's suggestion that we perform a scaling analysis of the k-N relationship. Our arguments regarding this point follow in the detailed response below. At a high level these arguments are the (1) the analysis has been performed elsewhere and we have cited the relevant literature, and (2) the smallest possible observable scale where the microphysical processes are occurring is the most relevant scale, and (3) it is not even possible to fill out a 2D area representative of a satellite footprint with a linear probe measurement.

**Response to Review #1**

This study uses in situ observations of droplet size distributions (DSD) and number concentrations (Nd) to investigate the relationship between DSD width (parameterized by the k parameter) and Nd. Generally, a positive k-Nd relationship is found. This relationship is parameterized and applied to Nd retrievals derived from MODIS level-2 cloud products.

The paper is well written and the subject is definitely relevant and interesting. However, at the end of the introduction the authors state that they claim to address the issue that "the spatial scale of the data is a critical determinant of the derived k-N relationship", which, in my view, they do not really do. Hence, I would only recommend publication if this issue is indeed addressed. I elaborate on this major comment below, after which some minor comments and suggestions are listed.

**Major comment:**

As said, the spatial scale on which the DSD is evaluated is critical for the k-Nd relationship. It seems that it is claimed that this issue is addressed by using high resolution (1Hz or ~100m) data. However, there is no demonstration on the scale dependency. More importantly, the resulting parameterization is then applied to correct satellite data that is on the order of 1km^2 resolution. The effective radius that is retrieved for a MODIS footprint would be the one that represents the 3rd over the 2nd moment of all drops near cloud top within that pixel. I would think that the k to derive Nd should the one that represents that MODIS resolution. If the mode (or effective) radii of DSDs at small scales over that footprint vary, this translates into an effectively wider total DSD for the whole footprint. So shouldn't the in situ observations (and thus derived k-Nd relations) be evaluated over a similar scale as the MODIS observations? It could be that a similar result is then obtained, but this needs to be shown. Or maybe the authors can convince me and the readers that the k-Nd relationship derived at fine scale is correctly applied to the coarser MODIS scales, but then I would argue for a discussion in the paper making this point.

We appreciate the comment. We believe the scale dependance of the k-N relationship is important and that it deserves significantly more study, which is why we state as much in the text. That said, we do not claim to evaluate the spatial scale dependence in this paper. Instead, we justify this point by including citations to four studies, three of which are observational, and one of which is theoretical, that suggest that the sign of the correlation tends to be positive at the microscale (< 2 km) but negligible at the meso-beta scale (20-200 km). Most satellite sensors have horizontal resolutions that fall in the microscale. Furthermore, it is not possible to perform the suggested analysis (e.g. evaluating the DSD at the resolution of MODIS footprint, since the footprint is a ~1km *square* area whereas the probe just represents a *linear* flight line. Even using a 1 km flight line would barely represent the variability that could occur in a 2D 1km2 area. In addition, we find it appealing to provide a parameterization of the relationship at the highest possible resolution, which is more representative of the scale of the actual microphysical processes, with the understanding that it could be useful for sensors of any resolution including those with potentially 100 m resolution in the future compared to legacy geostationary footprints of many square km. We have modified our discussion of this topic in several areas of the manuscript.

Our discussion in the introduction, which has been amended beginning on line 55 is repeated here:

Some of the disagreement in the reported k-N correlation can be explained by the spatiotemporal scale on which it is quantified. For example, Pawlowska et al. (2006) show that the relative dispersion of the DSD tends to decrease with N within a given flight leg (implying a positive k-N correlation), however the sign of this relationship reverses when evaluating flight averages. Hu et al. (2021) find no correlation for inter-cloud correlation either within campaigns or across the five campaigns they analysed. However, consistent with Pawloswka et al. (2006), they find when analysing the high-resolution 1 Hz DSDs that intra-cloud relative dispersion decreases with increasing number concentration. Hu et al. (2021) further show that the variance explained in the relative dispersion exceeds 40% up to scales of 10 km and quickly decreases at scales above 30 km. There is a theoretical basis for the scale dependence of the k-N correlation. An analytical model based purely on the condensational growth process (Liu et al., 2006) has demonstrated that for a fixed updraft velocity an increase in CCN concentration leads to an increase in N and an increase in relative dispersion, whereas for a fixed CCN an increase in updraft velocity leads to an increase in N but a decrease in relative dispersion. The former effect is interpreted to be relevant at larger scales (inter-cloud) while the later effect is interpreted to be relevant at smaller scales (intra-cloud). Data from aircraft penetrations of non-precipitating cumulus have confirmed the positive/negative correlation of updraft velocity with N/k respectively on the intra-cloud scale (Lu et al., 2012). The focus of this paper is on the problem of remote sensing, for which the relevant scale is variable depending on the sensor, but is generally within of the microscales (< 2 km). We call attention to the fact that many previous parameterizations of the k-N relationship are based on significantly larger spatial and temporal scales, which are likely not relevant to the remote sensing problem.

We have amended the discussion in the data and methods section (Line 132):

Here we evaluate 1 Hz data, which corresponds to length scale of approximately 100 m. It is acknowledged that the area footprint of a remotely sensed pixel is orders of magnitude larger than the linear sampling provided by the probes. However, it is not possible to reconstruct the two-dimensional area of a typical satellite pixel with probe data. The scale-analysis of Hu et al. (2021) shows that the DSD dispersion relationships at scales less than ~10 km have similar sign, so we choose to evaluate the data at the highest possible resolution and acknowledge that as with other aspects of the remote sensing problem, there is likely to be a pixel heterogeneity bias associated with non-linearities in the k-N relationship and the scale of the satellite footprint.

We have also added this comment in the summary and discussion at line 380:

We provide the k-N relationship at the highest possible spatial resolution with the understanding that this may introduce bias due to sub-pixel inhomogeneity. Knowledge of the sub-pixel distribution of N could be used to correct any such bias.

Note also that in response to reviewer #2 we have added the following paragraph in the summary and discussion, which relates to the implications of the scale dependance on climate simulations on Line 397:

'The emphasis of this paper is on the k-N correlation on the microscale ( $\leq 2$  km), which is relevant to the problem of remote sensing. Many previous studies have focused on the k-N correlation at the meso-beta scale (20-200 km) with a science focus on the parameterizing the aerosol indirect effect in climate models (e.g. Rostayn and Liu, 2003; Xie et al., 2017). While this study finds a positive k-N correlation, most studies exploring the aerosol indirect effect find or assume a negative correlation. Liu et al., 2006 has suggested that the positive microscale correlation is a result of variations in updraft velocity, whereas the negative mesoscale correlation is a result of variation in aerosol number concentration. Therefore, the correlations found here may not be applicable to evaluation of the aerosol indirect effect in climate simulations. We recommend that more work is necessary to fully resolve the scale-dependence of the k-N relationship and its implications for aerosol indirect effects.'

**Minor comments and suggestions:**

• Line 51: An interesting paper I suggest to add is the one by Sinclair et al. (GRL 2019; doi: 10.1029/2019GL085851), since they are using remote sensing rather than in situ to derive a relationship between DSD width (effective variance) and Nd. Using the conversion factor between effective variance and k from Grosvenor et al., the relationships Sinclair et al. find are somewhat comparable to the k-Nd relationship found in the current paper. Interestingly, they also find a strong relationship between LWP and k. Anyway, I leave it to the author's choice to discuss this paper and these results.

Thank you for pointing out this paper. We have added a reference to this paper at line 51.

'In addition to evidence from probe data analyses, retrievals of the droplet spectrum width and number concentration derived from the Research Scanning Polarimeter during the North Atlantic Aerosols and Marine Ecosystems Study show a positive k-N correlation (Sinclair et al., 2020).'

• Line 130: The threshold radius is rather specific and demands a reasoning or reference.

Assuming a threshold radius is never satisfying. Fortunately, the calculation of k has been shown to be very insensitive (< 2%) to the assumption of thresholds over the range (25-40 microns) that is commonly identified as the gap between cloud and precipitation modes. We have added the following text on Line 158 to justify our assumption:

This requires the definition of an arbitrary threshold radius separating the cloud and precipitation drops, which we choose as 27.5 µm. The calculation of k is not particularly sensitive to this threshold. For example, Brenguier et al. (2011) have shown that the calculation of k varies by less than 2% when varying the cloud/precipitation threshold between radii of 25 and 40 µm, which represents the range that is common to microphysical parameterizations in large eddy models (Geoffroy et al., 2010).

• Line 146: I am quite certain that the MODIS resolution is about 1km, so then 15 pixels would translate to 15 km. The 5 km might be a typo?

**Typo is corrected.**

• Figure 2: Interestingly, the histogram of k in panel D compares quite well with the histograms presented by Grosvenor et al (2018; their Fig 12b). Those are showing remote sensing data from the same instrument as discussed in the first minor comment. I suggest to point this out in the paper.

Great suggestion. We have added on line 205:

'The negatively skewed distribution, with values below 0.6 are not uncommon, which is consistent with the multi-angle polarimetric remote sensing derived distributions of k shown in Grosvenor et al., 2018 (their Figure 12)'

•Line 220, Eq 8: Is there a reason to use this particular functional form? Are there physical/theoretical interpretation of the k\_t, k\_b and N\* terms? I was confused at first about the subscripts B and T, which are often used for "Bottom" and "Top" of a cloud. I don't think this is the case here, so I would suggest using other subscripts.

We had intended b = bottom and t = top but for the DSD instead of the cloud physical heights as they are often used. To avoid confusion, we have changed these to k1 and k2. We have modified the following sentence (Line 266) to motivate the use of this functional form:

**'This functional form is ad-hoc, however it has two important properties: (1) it is bounded between 0 and 1, which is expected from physically realistic size distributions and (2) it increases monotonically and non-linearly while saturating for larger values of N, as we observe in the data'**

• Figure 6: As liquid water path can be quite well represented by 5/9 COT\*Reff, figure 6 would suggest that an implication of the parameterization is that there is a dependence of k on LWP as well. That might then be in agreement with the Sinclair et al. (2019) results. Or is this not a correct interpretation?

We do not think that Figure 6A supports this conclusion. In figure 6 isolines of LWP run from the top left to the bottom right with a slope of -1. Therefore, you can see that each value of LWP can be achieved with many combinations of tau and re that span orders of magnitude in N and therefore span the range of possible k. Furthermore, since this figure is based on our parameterization, which depends only on N, it is probably not appropriate to infer any dependance on LWP.

• Figure 7: I would suggest adding a figure of parameterized k to this figure if possible.

We have added a panel showing the mean value of the parameterized k.

• Line 282: Only here the scale of a 100 m is mentioned. Please mention this when first discussing the in situ data in section 2.

We have added at line 132:

'Here we evaluate 1 Hz data, which corresponds to length scale of approximately 100 m.'

**Response to Review #2**

This paper examines the dependence of the k-value (k=(rv/re)^3, where rv is volume mean radius and re is the effective radius) of cloud droplet size distributions upon the cloud droplet concentration (Nd) using aircraft data. A parameterization is developed to describe the dependency. This dependency leads to biases in the retrieval of Nd using MODIS retrieved re and cloud optical depth using the adiabatic cloud assumption since a constant k value is generally assumed for such retrievals. The biases due to assuming a constant k value of 0.8 are quantified using MODIS data subset to the CloudSat ground track (MAC06S0 dataset), presumably at ~1km resolution (although this is not clear from the text – see comments below). The science seems sound and the paper is very well written. The only thing missing is a little more discussion on how this might combine with some of the other potential biases relating to the precipitation, the k value and associated droplet distribution widths, and some discussion on the implications for climate models (which relates to the behaviour at larger spatial scales).

**Major points:**

It would be good to discuss the effect of k variability and the presence of precipitation on the MODIS retrievals of re. MODIS assumes a modified gamma distribution for cloud droplets for the radiative models used for its retrievals of re. The variation of k will also affect this and may lead to biases in re that would be likely to have a strong influence on Nd retrievals. The presence of a precipitation mode might also affect the retrieved re and hence Nd retrievals. I don't expect you to investigate the effect of this, but it would be worth mentioning in the discussion perhaps. Correcting for this bias could therefore either enhance or reduce the bias discussed in the paper being reviewed. Section 2.4.4. of Grosvenor et al. (2018) contains some useful discussion on this. From the information given there it seems likely that in precipitating regions neglecting the rain mode is likely to lead to an underestimate of Nd. These are likely to be regions with broader droplet distributions (low k) and the presence of a precipitation mode may therefore enhance the error seen in your paper since you also show an underestimate of Nd in such regions. In regions of high k the assumption by the MODIS retrieval of k=0.72 is likely to lead to an underestimate of Nd due to the DSD width assumption in the re retrieval, whereas your paper shows a positive Nd bias in such regions.

Zhang, Z. B., Ackerman, A. S., Feingold, G., Platnick, S., Pincus, R., & Xue, H. W. (2012). Effects of cloud horizontal inhomogeneity and drizzle on remote sensing of cloud droplet

effective radius: Case studies based on large-eddy simulations. Journal of Geophysical Research, 117, D19208. https://doi.org/10.1029/2012JD017655

Zhang, Z. (2013). On the sensitivity of cloud effective radius retrieval based on spectral method to bimodal droplet size distribution: A semi-analytical model. Journal of Quantitative Spectroscopy & Radiative Transfer, 129, 79–88. https://doi.org/10.1016/j.jqsrt.2013.05.033

These papers might be useful for the discussion of drizzle effects too at line 70.

This is a great point. We have added a paragraph including the Zhang (2013) reference discussing the inconsistencies between the MODIS retrieval and the derivation of N in the Summary and Discussion (Line 384):

'While it is straightforward to implement the parameterization presented here for the derivation of N from existing  $\tau$  and  $r_e$  retrievals, it is important to realize that there is a physical inconsistency between the parameterized k and the k assumed in the original retrieval of  $\tau$  and  $r_e$ . The MODIS retrievals assume a constant value of k = 0.72 (Grosvenor, 2018). It has been shown that bias in the retrieved  $r_e$  is inversely related to the bias in k (Zhang, 2013). That is to say that if the assumed k in the  $\tau/r_e$  retrieval is too large than the retrieved  $r_e$  will be too small. An implication of this work is that the MODIS re may contain a bias related to the relationship between k and N. Since the average k in the probe data is 0.8, on average this will bias MODIS *r*e low and the resultant N high, however the opposite would be true for very low N clouds. One solution to this inconsistency would be to build  $\tau/r_e$  look up tables using an adiabatic model with input N, cloud thickness, and condensation rates in a manner similar to Schüller et al., (2003), while also including the k-N parameterization used in this paper. One could further increase the realism of the look up tables by including a parameterization of the cloud subadiabaticity as a function of cloud thickness as in Schulte et al. (2023). In this manner a complete and physically consistent description of the cloud microphysical profile can be constructed from the initial inversion of the radiances.'

With regard to precipitation, we have cited the Zhang 2012 paper and added the following sentence at line 166:

**'The assumption that the radiative effects of drizzle mode drops do not contribute to the derived cloud top re is consistent with the simulation study of Zhang et al. (2012).'**

Some climate models use the parameterization of Liu (Y. Liu, P. H. Daum, H. Guo, and Y. Peng. Dispersion bias, dispersion effect and the aerosol-cloud conundrum. Environ. Res. Lett., 3:045021, 2008.), and other similar papers from those authours. For example the UK Earth System Model (UKESM) as described in Mulcahy (2018;

https://agupubs.onlinelibrary.wiley.com/doi/epdf/10.1029/2018MS001464). It would be good to discuss the fact that this could also be important for climate models in the introduction. It would also be useful to discuss what the implications of the results of your paper are for such climate models – this would probably relate to the issue of the scale dependence. The Liu and Daum parameterization has the opposite dependence of k on N to your parameterization, although the

scales of a climate model grid box are much larger (1x1 degree or so, so around 100km). The impact may depend on the details of how SW radiative fluxes are calculated in the climate models (e.g., using grid-box wide values or sub columns, etc.).

This is an important discrepancy to point out. We don't want to confuse the focus of the readers in the introduction. Therefore, we have added a final paragraph in the summary and discussion (Line 397) that attempts to caution readers about the importance of spatial scale and causal mechanism in determining the k-N correlation.

'The emphasis of this paper is on the k-N correlation on the microscale ( $\leq 2$  km), which is relevant to the problem of remote sensing. Many previous studies have focused on the k-N correlation at the meso-beta scale (20-200 km) with a science focus on the parameterizing the aerosol indirect effect in climate models (e.g. Rostayn and Liu, 2003; Xie et al., 2017). While this study finds a positive k-N correlation, most studies exploring the aerosol indirect effect find or assume a negative correlation. Liu et al., 2006 has suggested that the positive microscale correlation is a result of variations in updraft velocity, whereas the negative mesoscale correlation is a result of variation in aerosol number concentration. Therefore, the correlations found here may not be applicable to evaluation of the aerosol indirect effect in climate simulations. We recommend that more work is necessary to fully resolve the scale-dependence of the k-N relationship and its implications for aerosol indirect effects.'

**Minor points**

Eqns.3 to 6 – you should probably make it clear that you are summing over the different size bins of the droplet probes with subscript i values for the bin quantities  $(n(r), r, \Delta r)$  and you should say what n(r), rmin and rmax refer to. Also, Eqn. 6 switches to using a capital N for n(r).

We have defined n(r), rmin and rmax. We have also corrected N -> n in Eqn. 6. We choose not to include the subscript i, which should be intuitive given the use of a summation operation. This text has been modified on line 156:

'Here n(r) is the discretely binned droplet size distribution and  $\Delta r$  is the bin width.  $r_{min}$  and  $r_{max}$  are the bounds of the summation. Each quantity is calculated three times with different min/max values, for the cloud mode, precipitation mode, and total DSD.'

 $L146 - "15 \text{ pixel} (\pm 5 \text{ km}) \text{ swath}" - \text{this is a bit confusing since MODIS pixels for re at least are usually 1km across. Should it be 10 pixels, or <math>\pm 7.5 \text{ km}$ ? Otherwise it would be good to explain this in the text. You should also say what the resolution of the MODIS data that you are using is.

This was a typo. ( $\pm$  5 km) has been changed to ( $\sim$ 15 km) on Line 178. We have also identified that it is Level-2 data to identify the 1 km scale.

L225 – since you reduce the weighting of the campaigns with more sample points in order to get more equal representation between the campaigns, would a straight average of the three curves not achieve something similar?

The method that we chose minimized the weighted difference between the data and the parameterized curve (in a least squares sense). For this reason we consider it the optimal approach. We don't think that a straight average of the three curves is the correct approach. If we did that we would then need to take that average curve and refit for the parameters k1, k2, and N\*. The end result might be similar, but it is a somewhat awkward approach.

Fig. 5 - it seems like a 2d histogram would be better than the scatter points (or contours with different colours for each campaign if wanting to separate them). You should also say what the scatter points are in the caption.

The data is presented in a similar fashion to what is suggested in Figure 4 with the box whisker plots. The point of Figure 6 is to show the parameterized curves. We only include the scattered data points for illustrative purposes. We choose to keep this figure as it is.

Typos

L45 – Heymfield -> Heymsfield L69 - that collision coalescence process -> that the collision coalescence process 179 - at low values N -> at low values of N L86 - measurements of drops size distribution -> measurements of droplet size distributions L268 - difference -> differences

Thank you. All of these are corrected.